# Characterization of Two Unique Cold-Active Lipases Derived from a Novel Deep-Sea Cold Seep Bacterium

**DOI:** 10.3390/microorganisms9040802

**Published:** 2021-04-10

**Authors:** Chenchen Guo, Rikuan Zheng, Ruining Cai, Chaomin Sun, Shimei Wu

**Affiliations:** 1College of Life Sciences, Qingdao University, Qingdao 266071, China; 2018025209@qdu.edu.cn; 2CAS Key Laboratory of Experimental Marine Biology & Center of Deep Sea Research, Institute of Oceanology, Chinese Academy of Sciences, Qingdao 266071, China; zhengrikuan15@mails.ucas.ac.cn (R.Z.); ruiningcai@163.com (R.C.); 3Laboratory for Marine Biology and Biotechnology, Qingdao National Laboratory for Marine Science and Technology, Qingdao 266071, China; 4Center of Ocean Mega-Science, Chinese Academy of Sciences, Qingdao 266071, China

**Keywords:** lipase, cold seep, cold-active, deep sea, overexpression

## Abstract

The deep ocean microbiota has unexplored potential to provide enzymes with unique characteristics. In order to obtain cold-active lipases, bacterial strains isolated from the sediment of the deep-sea cold seep were screened, and a novel strain gcc21 exhibited a high lipase catalytic activity, even at the low temperature of 4 °C. The strain gcc21 was identified and proposed to represent a new species of *Pseudomonas* according to its physiological, biochemical, and genomic characteristics; it was named *Pseudomonas marinensis*. Two novel encoding genes for cold-active lipases (Lipase 1 and Lipase 2) were identified in the genome of strain gcc21. Genes encoding Lipase 1 and Lipase 2 were respectively cloned and overexpressed in *E. coli* cells, and corresponding lipases were further purified and characterized. Both Lipase 1 and Lipase 2 showed an optimal catalytic temperature at 4 °C, which is much lower than those of most reported cold-active lipases, but the activity and stability of Lipase 2 were much higher than those of Lipase 1 under different tested pHs and temperatures. In addition, Lipase 2 was more stable than Lipase 1 when treated with different metal ions, detergents, potential inhibitors, and organic solvents. In a combination of mutation and activity assays, catalytic triads of Ser, Asp, and His in Lipase 1 and Lipase 2 were demonstrated to be essential for maintaining enzyme activity. Phylogenetic analysis showed that both Lipase 1 and Lipase 2 belonged to lipase family III. Overall, our results indicate that deep-sea cold seep is a rich source for novel bacterial species that produce potentially unique cold-active enzymes.

## 1. Introduction

Cold environments such as those in the deep sea [1], glaciers, and mountain regions are one of the most abundant environments for microorganisms on the Earth’s surface. Even in the cold environments, where the temperatures are close to 5 °C, these regions are still colonized successfully by numerous microorganisms [2]. In order to cope with the harsh effects of such environments, these microorganisms have evolved various adaptation strategies, such as producing enzymes with high specific activity at low temperatures, collectively termed cold-active enzymes [3]. Cold-active enzymes usually have high catalytic activities at temperatures below 25 °C, which gives them great advantages in detergent, textile, and food industries because of the energy savings they provide [4]. Furthermore, efficient catalysis at low temperatures of the cold-active enzyme requires an increase in protein flexibility, while high flexibility is also accompanied by a trade-off in thermal stability [5]. These characteristics of cold-active enzymes not only prevent adverse reactions at higher temperatures, but also provide a mild way for their rapid thermal inactivation due to their low thermal stability, which is particularly significant in the food industry as it avoids the use of chemical-based inactivation and prevents the damage of food nutrition [6].

Lipases constitute the third most important category of enzymes, next to carbohydrases and proteases [7]; they can act on the carboxyl ester bonds of acylglycerols to produce fatty acids and glycerols [8] and are widely applied in various industries. The global lipase market is estimated to reach USD 797.7 million by 2025 at a compound annual growth rate at 6.2% from 2017 to 2025 [9]. Compared with lipases derived from plants or animals, the microbial lipases are more attractive due to the easy culture of microorganisms, high yield production, and simplicity of genetic manipulation [10]. Recently, lipases produced by cold-adapted bacteria have attracted much attention because of their low optimum temperatures and high catalytic activities. A number of cold-active lipases have been reported so far, and most of the lipase-producing strains, including strains from the genera *Pseudoalteromonas*, *Pseudomonas*, *Psychrobacter*, *Photobacterium*, and *Colwellia*, were isolated from polar regions including Antarctica and deep-sea environments [11]. These cold-active lipases show high specific activities in the temperature range of 0–30 °C and exhibit promising potentials in detergent additives, leather, bioremediation, food processing, and pharmaceutical industries due to the consumption of less energy [12,13].

In this study, in order to obtain cold-active lipases, bacteria were isolated from the sediment of the deep-sea cold seep in the South China Sea, and their lipase-producing activities were detected. A novel strain exhibited the highest lipase activity at low temperature, even at the temperature of 4 °C, and then the lipase-producing strain was identified and named *Pseudomonas marinensis* gcc21 according to the results of its genome sequencing and biological characteristics. Furthermore, the genes of strain *P. marinensis* gcc21 encoding two cold-active lipases were cloned and overexpressed in *Escherichia coli* cells and then purified and biochemically characterized.

## 2. Materials and Methods

### 2.1. Screening of Lipase-Producing Bacteria from Deep-Sea Cold Seep Samples

The deep-sea sedimental samples were collected at a depth of 1146 m in the cold seep (22°06′58.598″ N 119°17′07.322″ E), P.R. China, in June 2018, and bacteria from the samples were isolated on the modified Zobell 2216E medium (5.0 g peptone, 1.0 g yeast extract, 15.0 g agar, a liter of filtered seawater, pH 7.0) at 28 °C as described previously [14,15]. To screen lipase-producing bacteria, isolated strains were cultivated in plates containing 2216E medium supplemented with 1% (*v*/*v*) Tween 80 [16]. The plates were incubated at 28 °C for 48 h, and bacteria producing an obvious halo-forming zone were evaluated as potential lipase-producing strains. The isolated strains were streaked three times on the same medium to ensure their purity and preserved in 2216E liquid medium (5.0 g peptone, 1.0 g yeast extract, a liter of filtered seawater, pH 7.0) supplemented with 20% (*v*/*v*) glycerol at −80 °C. The novel lipase-producing stain gcc21 was deposited in the China General Microbiological Culture Collection Center under collection number CGMCC 1.18552.

### 2.2. Physiological Characterizations of Strain gcc21

To detect the physiological characteristics of strain gcc21, its closely related strains *P. sabulinigri* J64^T^ and *P. aeruginosa* DSM50071^T^ were used as reference strains in most of the subsequent phenotypic tests and grown on 2216E under the same conditions. The morphological characteristics of strain gcc21 were observed under TEM (HT7700; Hitachi, Tokyo, Japan). The temperature range of the strain growth assay was tested at different temperatures (4 °C, 16 °C, 28 °C, 30 °C, 37 °C, 45 °C, 60 °C, 70 °C, 80 °C) for 5 days in 2216E liquid broth. The pH range of strain growth was tested in the 2216E liquid broth from 2.0 to 10.0 with increments of 0.5 pH units and incubated at 28 °C for 5 days. Salt tolerance was tested in the modified 2216E liquid broth (distilled water replaced sea water) supplemented with 0–10% (*w*/*v*) NaCl (0.5% intervals) at 28 °C for 5 days. Catalase activity was evaluated by observation of bubbles production in fresh bacterial solution with 3% (*v*/*v*) H_2_O_2_ solution. Oxidase activity was determined using the oxidation of 1% (*w*/*v*) tetramethyl p-phenylenediamine to observe the color change of reaction solution [17]. Hydrolytic capacities of Tween 20 and Tween 80 were tested by observing the diameter of the halo-forming zone on the solid 2216E medium containing 1% (*w*/*v*) test substances at 28 °C for 5 days [16]. Hydrolysis ability of starch was determined by gram-iodine on colonies on the solid 2216E medium containing 1% (*w*/*v*) starch at 28 °C for 5 days. Substrate utilization was tested in the medium containing sterile seawater and 0.002% (*w*/*v*) yeast extract supplemented with a single substrate at a final concentration of 20 mM, such as urea, glucose, acetate, maltose, butyrate, fructose, glycine, ethanol, formate, lactate, sucrose, L-rhamnose, insotiol, L-arabitol, xylitol, sorbitol, D-mannose, and D-mannitol. Cell culture containing only yeast extract without any other substrates was used as a control. All media were adjusted to pH 7.0 with NaOH or HCl. The cultures were incubated at 28 °C in a shaker with a speed of 150 rpm for 5 days, and the absorbance was measured at 600 nm (A600). Each experiment was repeated three times. Cells respiratory quinones were extracted and analyzed by HPLC (model 1200, Agilent, Santa Clara, CA, USA) according to the methods described previously [18,19]. In addition, cellular fatty acids were extracted and determined by using GC (model 7890A, Agilent, Santa Clara, CA, USA) according to the protocol of the Sherlock Microbial Identification System [20], and polar lipids were extracted and identified as described by Tindall et al. [21]. 

### 2.3. Phylogenetic Analysis

Genomic DNA was extracted by using a bacterial genomic kit (TIANGEN, Beijing, China). The sequence of the 16S rRNA gene was amplified by PCR with primers 27-F and 1492-R (Table 1) [22]. PCR was performed using a 20 μL reaction mixture as follows (per reaction): 10 μL 2X Rapid Taq Master Mix (Vazyme, Nanjing, China), 9 μL of sterilized water, 0.5 μL of each primer (10 mM), and 1 μL of template DNA. The purified PCR products were sequenced by the Sanger sequencing method [23]. Phylogenetic trees were constructed with the full-length 16S rRNA sequences by the neighbor-joining algorithm [24], maximum likelihood [25], and minimum-evolution methods [26]. 16S rRNA sequences of other related strains used for phylogenetic analysis were obtained from NCBI GenBank. Phylogenetic analysis was performed using the software MEGA version 6.0 [27].

### 2.4. Genomic Characterizations of Strain gcc21

The whole genome sequencing (WGS) of strain gcc21 was performed with Oxford Nanopore MinION sequencing technology and assembled by Canu v1.5 [28]. Genome relatedness values were calculated by multiple approaches: ANIm and ANIb were calculated by using JSpecies WS (http://jspecies.ribohost.com/jspeciesws/, accessed on 28 February 2021) [29] and *Is*DDH values were calculated based on the recommended formula 2 by the Genome-to-Genome Distance Calculator (GGDC) (http://ggdc.dsmz.de/, accessed on 28 February 2021) [30].

### 2.5. Overexpression, Gene Mutagenesis, and Purification of the Cold-Active Lipases in E. coli

The genes encoding Lipase 1 and Lipase 2 were amplified by PCR and inserted into the restriction sites of expression plasmid pET-28a (+). Lipase 1 was cloned into pET-28a (+) with restriction endonuclease *EcoR* I and *Xho* I, and Lipase 2 was cloned into pET-28a (+) with restriction endonuclease *Hind* III and *Bam* HI, respectively. PCR was performed using a 40 μL reaction mixture (per reaction) that included 20 μL 2X KOD One^TM^ PCR master Mix (Vazyme, Nanjing, China), 14 μL of sterilized water, 2 μL of each primer (10 mM), and 2 μL of template DNA, and the follow procedure was followed for 36 cycles: denaturation at 98 °C for 15 s, annealing at 60 °C for 10 s, extension at 68 °C for 5 s, and final extension at 68 °C for 5 min. Meanwhile, by using recombinant plasmids as templates, mutants of Lipase 1 (S151A, D201A, and H231A) and mutants of Lipase 2 (S160A, D210A, and H240A) were created with the KOD-Plus-Mutagenesis kit with primers containing mutation sites as listed in Table 1. All the recombinant plasmids were verified by Sanger sequencing method [23]. The resulting plasmids were transformed into *E.coli* BL21 (DE3) cells, and incubated in Luria-Bertani medium (LB) (10 g tryptone, 10 g NaCl, 5 g yeast extract, one liter of distilled water, pH 7.0) with kanamycin (100 μg/mL) at 37 °C. The corresponding transformant was cultured in LB broth with kanamycin (100 μg/mL) at 37 °C until the absorbance at 600 nm (A_600_) reached about 0.5, then isopropyl-beta-D-thiogalactopyranoside (IPTG) with a final concentration of 0.2 mM was added into the culture medium to induce protein expression at 16 °C for 20 h. Subsequently, the culture was centrifuged at 8000× *g* for 20 min to collect the cells. Cell pellets were resuspended with 150 mM NaCl containing 20 mM Tris-HCl (pH 8.0) and disrupted by sonication and then centrifugated at 8000× *g* for 30 min to obtain the supernatant. The purification of enzymes was performed by using a Ni-NTA resin column (GE Healthcare, Chicago, IL, USA) and a gradient elution with 500 mM imidazole containing 150 mM NaCl and 20 mM Tris-HCl (pH 8.0) on an AKTA protein purification system (GE Healthcare, USA), and the eluent was collected throughout the process. After that, the components of active peaks were concentrated by ultra-filtration (3 kDa MW interception membrane, Millipore, USA) and subjected to the Hiload^TM^ 16/600 Superdex^TM^ 200 column (GE Healthcare, Boston, MA, USA) for gel filtration. A molecular sieve was equilibrated with 150 mM NaCl containing 20 mM Tris-HCl (pH 8.0) and used to collect the active components. All processes were performed at 4 °C. The purity of the enzymes was examined by 10% polyacrylamide gel electrophoresis in SDS-PAGE. Total protein concentration was calculated using a nanophotometer (Implen, Munich, Germany).

### 2.6. Sequence Analysis

Lipase and hydrolase sequences for comparative study were obtained from protein and nucleotide databases on NCBI (http://www.ncbi.nlm.nih.gov/Entrez/, accessed on 28 February 2021) and amino acid sequences were aligned by CLUSTALW [31]. Phylogenetic trees were constructed by the neighbor-joining algorithm with the software MEGA version 6.0. Bootstrap values at nodes were derived from 1000 replicates [32].

### 2.7. Enzyme Assays

Lipase activity was carried out by using *p*-nitrophenyl palmitate (C16, *p*-NPP) as a substrate unless otherwise indicated [33]. The reaction mixture consisted of 0.1 mL enzyme extract (5 μg/mL), 0.8 mL of 50 mM Tris-HCl (pH 8.0), and 0.1 mL of 10 mM *p*-NPP dissolved in isopropyl alcohol. The reaction mixture was incubated at 4 °C for 30 min, then stopped by adding 0.25 mL of 100 mM sodium carbonate and centrifugated at 10,000× *g* for 10 min. Absorbance of the supernatant was measured at 410 nm using a microplate reader (Tecan, Switzerland). One unit of lipase was defined as the amount of enzyme required to liberate 1 µmol of *p*-nitrophenol per minute from *p*-NPP [34]. Total protein concentration was calculated using a nanophotometer (Implen, Munich, Germany). To detect the lipase activity towards various substrates, *p*-nitrophenyl acetate (C2), *p*-nitrophenyl butyrate (C4), and *p*-nitrophenyl laurate (C12) were selected as substrates. All experiments were performed under the same conditions and repeated three times. The non-enzymatic hydrolysis factors of the substrates were removed by using the reaction system without the enzyme as a control.

### 2.8. Effects of Temperature and pH on the Lipase Activity and Stability

The optimum temperature and pH for lipase activity were determined with *p*-NPP as a substrate. The assay was performed by incubation of the reaction mixture at various temperatures and pHs for 30 min. To investigate the temperature stability, the lipase solution was pre-incubated for 1 h at different temperatures ranging from 4 °C to 80 °C. For the pH stability assay, the lipase solution was pre-incubated for 1 h at different pH values with the buffer system as follows: 0.1 M citric acid/sodium citrate buffer (pH 5), 0.1 M potassium phosphate buffer (pH 6–7), 0.1 M Tris-HCl (pH 8), and 0.1 M glycine-NaOH (pH 9–12) with intervals of 1.0 pH unit at room temperature. All residual activity was measured by the methods mentioned above. Each experiment was repeated three times.

### 2.9. Effects of Metal Ions, Inhibitors, Detergents, and Organic Solvents on Lipase Activities

The lipase solutions were pre-incubated with different concentrations of various metal ions, enzyme inhibitors, detergents, and organic solvents for 1 h at room temperature, and the residual lipase activities were determined as described above. The enzyme solution without any treatment was considered as a control, and each experiment was repeated three times.

### 2.10. Statistical Analysis

The quantitative data were expressed by using mean ± standard deviation. The differences among results were analyzed by one-way analysis of variance (ANOVA) based on Turkey’s test and the least significant difference (LSD) test using SPSS 11.0 statistical software. Statistical significance was considered as *p* value < 0. 05.

### 2.11. Data Availability

The 16S rRNA gene sequence and whole-genome sequence (WGS) of *P. marinensis* gcc21 were deposited in the Gene Bank database under accession numbers MT560371 and CP051625, respectively. The complete amino acid sequences of the Lipase 1 and Lipase 2 were deposited in the GenBank database under accession numbers WP_169406187.1 and WP_169408678.1, respectively, and corresponding DNA sequences of Lipase 1 and Lipase 2 were deposited in the GenBank database under accession numbers MW822015 and MW822016, respectively.

## 3. Results

### 3.1. Physiological Characterization of Lipase-Producing Strain gcc21 Isolated from the Deep-Sea Cold Seep

After lipase-producing screening, a novel strain gcc21 exhibited desirable lipase activity even at the low temperature of 4 °C. The 16S rRNA gene of strain gcc21 was sequenced and showed the highest identity of 97.7% with that of *Pseudomonas salina* XCD-85^T^, and 97.2% with that of *Pseudomonas litoralis* 2SM5^T^, indicating that strain gcc21 maybe represent a novel species of the genus *Pseudomonas*.

Strain gcc21 was a Gram-negative, rod-shaped bacterium, 0.8–1.0 µm wide and 1.1–1.8 µm long, as shown by transmission electronic microscopy (TEM) (Appendix A). As indicated in Table 2, the strain was able to grow in a temperature range from 4 to 37 °C (optimum, 28 °C), a pH ranging from 5.0 to 8.5 (optimum, pH 7.0), and a NaCl concentration of 0.5 to 9.0% (*w*/*v*) (optimum, 1.5%). In addition, strain gcc21 exhibited both oxidase and catalase activities, and could hydrolyze Tween 20 and Tween 80, but could not hydrolyze starch. It could utilize ethanol, sorbitol, D-mannose, L-arabitol, xylitol, acetate, and lactate, but could not use urea, glucose, maltose, butyrate, fructose, glycine, formate, sucrose, L-rhamnose, insotiol, or D-mannitol. The major polar lipids of strain gcc21 were diphosphatidylglycerol (DPG), phosphatidylglycerol (PG), phosphatidylethanolamine (PE), unidentified phospholipid (PL), and an unidentified aminophospholipid (APL) (Appendix A). The predominant respiratory quinone of strain gcc21 was ubiquinone-9 (Q-9), typical in the family Pseudomonadaceae [35]. The genomic DNA G+C content was 58.27%, which is at the limit of the range of 58–69% known for the genus *Pseudomonas* [36]. The predominant fatty acids were branched-C_17:0_ cyclo, branched-C_16:0_, summed feature 3, and summed feature 8. As shown in Appendix A, the amounts of branched-C_17:0_ cyclo (16.26%), summed feature 3 (15.99%) and summed feature 8 (26.87%) in strain gcc21 were much higher than those in *Pseudomonas aeruginosa* DSM50071^T^ (9.13%, 4.77%, and 18.97%, respectively), while the amount of branched-C_16:0_ in strain gcc21 (17.38%) was lower than that in *P. aeruginosa* DSM50071^T^ (31.61%). The results of ANIm, ANIb between strain gcc21, and the relative available genome of *Pseudomonas* strains were significantly below the proposed cut-off for a species boundary of 95–96% of DNA similarity [37], and the *is*DDH values were clearly below the proposed thresholds for prokaryotic species delineation at 70% [38], which together support the notion that strain gcc21 represented a distinct species (Appendix A). Meanwhile, the phylogenetic tree was constructed by the neighbor-joining algorithm (Figure 1), maximum likelihood (Appendix A), and minimum-evolution (Appendix A) methods, which indicated that strain gcc21 failed into the clade that contained *Pseudomonas* species, forming a cluster with *P. litoralis* 2SM5^T^, *P. formosensis* CC-CY503^T^, and *P. xiamenensis* C10-2^T^. All of the above results indicated that strain gcc21 could represent a novel species of the genus *Pseudomonas,* and we proposed the name *Pseudomonas marinensis*. The strain type is *Pseudomonas marinensis* gcc21^T^.

### 3.2. Overexpression and Purification of the Cold-Active Lipases in E. coli

Given that strain gcc21 showed a great lipase activity at the low temperature of 4 °C, we thoroughly analyzed the genome of strain gcc21. Indeed, two encoding genes for cold-active lipase were found in the genome of strain gcc21, and their encoded proteins were named Lipase 1 and Lipase 2 with a calculated molecular mass weight of 29.44 kDa and 30.35 kDa, respectively. Lipase 1 showed the highest identity of 65.4% with a lipase derived from *Pseudomonas bauzanensis,* W13Z2^T^, and Lipase 2 showed the highest identity of 65.1% with a lipase derived from *Pseudomonas gallaeciensis,* V113^T^. The homology between Lipase 1 and Lipase 2 was 71.4%. In order to investigate the biochemical properties of Lipase 1 and Lipase 2, corresponding genes were cloned and overexpressed in *E. coli* BL21 cells. The single elution peaks of Lipase 1 and Lipase 2 were obtained at the final purification step (Figure 2A). The purified active fractions were detected by sodium dodecyl sulfate-polyacrylamide gel electrophoresis (SDS-PAGE) and corresponding single protein bands around 30 kDa for both Lipase 1 and Lipase 2 were seen in the gel (Figure 2B), indicating that their purities were high enough for further biochemical assays.

### 3.3. Characterizations of the Cold-Active Lipases

To determine the optimal pH and temperature for Lipase 1 and Lipase 2, the activities of purified Lipase 1 and Lipase 2 at various pHs and temperatures were detected. Lipase 1 showed a different extent of activity over a wide pH range, from 5.0 to 11.0, with the highest activity at the pH 8.0. This Lipase also exhibited the highest stability at pH 8.0 (Figure 3A). Lipase 2 exhibited similar optimal pH activity and stability but showed higher activity and stability (Figure 3B). Lipase 1 exhibited the highest activity and stability at the lowest temperature tested (4 °C), while the activity and stability decreased sharply with increasing temperature (Figure 3C). Lipase 2 showed similar activity and stability, however, these gradually decreased with increasing temperature (Figure 3D).

Next, stabilities of Lipase 1 and Lipase 2 against different metal ions, detergents, potential inhibitors, and organic solvents were assayed. For the stability detection against different ions (Figure 4A), K^+^ caused an increase in the activity of Lipase 1 and had little effect on Lipase 2, while Mg^2+^ caused an increase in the activity of Lipase 2 and had little effect on Lipase 1 (Figure 4A). The activities of both Lipase 1 and Lipase 2 were not affected intensely by Ba^2+^ and Ca^2+^, but dropped off by different degrees with Cu^2+^, Mn^2+^, and Zn^2+^, except for the slight effect of 1 mM Cu^2+^ on Lipase 2 (Figure 4A). In addition, the stabilities of Lipase 1 and Lipase 2 were strongly reduced by most of detergents (Figure 4B) and most inhibitors, such as thiourea, DTT, and urea (Figure 4C). However, the activity of Lipase 2 was not severely affected by inhibitors EDTA and PMSF. For stability assays against different organic solvents (Figure 4D), ethanol and glycerin had a slight effect on Lipase 2, while xylene, N-butanol, DMSO, methanol, and P-xylene greatly reduced the activity of Lipase 2, and the activity of Lipase 1 was strongly reduced by most of the organic solvents.

In order to detect the optimal substrate of Lipase 1 and Lipase 2, the enzyme activity towards various *p*-nitrophenyl esters was examined at 4 °C. As shown in Figure 5, both Lipase 1 and Lipase 2 exhibited the maximum activity against C12, but Lipase 2 showed a higher activity than that of Lipase 1 against C16. The kinetic parameters of Lipase 1 and Lipase 2 were also tested with *p*-NPP as a substrate. As shown in Table 3, the K_m_ is 0.02 for Lipase 1 and 0.06 for Lipase 2. The V_max_, K_cat_, and K_cat_/K_m_ of Lipase 2 are much higher than those of Lipase 1, which further confirmed that the activity of Lipase 2 is much higher than that of Lipase 1.

### 3.4. Functional Verification of Key Amino Acids and Phylogenetic Analysis of Lipase 1 and Lipase 2

To gain deeper insights into Lipase 1 and Lipase 2, their amino acid sequences together with those of their homologs were analyzed. Multiple sequence alignment suggested that the catalytic triads of Lipase 1 and Lipase 2 were composed of residues Ser151-Asp201-His231 and Ser160-Asp210-His240, respectively (Figure 6), and the composition of the catalytic triad containing Ser-Asp-His is a typical characteristic of the α/β-hydrolase superfamily [41]. In addition, the catalytic sites Ser151 of Lipase 1 and Ser160 of Lipase 2 are located in the typical GxSxG motif, which is one of the known characteristics of lipolytic enzymes [42]. In order to confirm the importance of these catalytic sites, directed mutagenesis was carried out on these residues. Accordingly, all the mutants showed expression levels and purities as good as those of wild type (Figure 7A). As expected, mutation of these residues to Ala completely resulted in inactivation of Lipase 1 and Lipase 2 (Figure 7B), thus demonstrating the key role of residues Ser, Asp, and His in Lipase 1 and in Lipase 2 as reported in other lipases [43].

In order to investigate the category of Lipase 1 and Lipase 2, a phylogenetic tree was established according to the amino acid sequences of Lipase 1 and Lipase 2 and other reported lipases (Figure 8), the result showed that Lipase 1 and Lipase 2 were more phylogenetically related to the family III, which displays the canonical fold of α/β-hydrolases and contains a typical catalytic triad.

## 4. Discussion

The *Pseudomonas* genus represents a kind of Gram-negative, aerobic bacteria, which belongs to the family Pseudomonadaceae of class Gammaproteobacteria [44]. It is considered to be a common member of microbial communities, occupying various environments on Earth. A large number of novel *Pseudomonas* species have been isolated from different environments such as soil, fresh and saline waters, plants, and clinical specimens [45], while relatively few species have been reported from the deep-sea cold seep. In the current study, a lipase-producing *Pseudomonas* strain gcc21 was isolated from the deep-sea cold seep, and the highest 16S rRNA sequence homology with other *Pseudomonas* strains was 97.7%, indicating that strain gcc21 might be a novel *Pseudomonas* strain. Further study showed that strain gcc21 had the typical respiratory quinone (Q9) of *Pseudomonas* [35], and exhibited differences from representative *Pseudomonas* strains in major fatty acids, polar lipids, ANI and *is*DDH values, and physiological and biochemical characteristics. Therefore, our results showed that strain gcc21 could represent a novel species of the genus *Pseudomonas*, and the strain was named *Pseudomonas marinensis.* In addition, though strain *P. marinensis* gcc21 was isolated from the deep-sea cold seep, the strain grew much better at 28 °C than at low temperatures, and the strain had a low utilization rate of glucose, just as previously report for *Pseudomonas profundi* sp. nov., which was isolated from deep-sea water and exhibited the optimal growth temperature at 25 °C and was negative for glucose fermentation and assimilation [44]. Therefore, our results indicate that the deep sea is an important source from which to isolate new species.

Since *P. marinensis* gcc21 exhibited the ability to produce lipase, two lipases (Lipase 1 and Lipase 2) from this strain were cloned and expressed in *E. coli* BL21. Further study showed that Lipase 1 and Lipase 2 from *P. marinensis* gcc21 were two novel cold-active lipases with an optimal catalytic temperature at 4 °C among the temperatures we tested, similar to that for the lipase from a pychrotrophic bacterium *Pseudomonas* sp. PF 16^T^ [46]. The optimal catalytic temperature of Lipase 1 and Lipase 2 was much lower than that of most reported cold-active lipases [47,48,49,50,51,52,53,54,55], which indicates that the deep-sea cold seep is an ideal source to screen novel cold-active enzymes. Notably, most of the cold-active lipases exhibit high catalytic activities at low temperature but have a trade-off of low stability [46,56]. In our study, Lipase 2 exhibited a relatively high stability, which allowed it to maintain more than half of its maximum activity in a temperature range of 4–40 °C, and pH range of 6.0–11.0. Therefore, Lipase 2 is a special cold-active lipase with high stability, strongly suggesting that Lipase 2 may be useful in future applications.

Based on the conserved sequence motifs and the biological properties of esterase/lipases, bacterial esterase/lipases were divided into eight different families [57], and the phylogenetic tree analysis showed both Lipase 1 and Lipase 2 belonged to family III. In addition, Lipase 1 and Lipase 2 showed a high sequence similarity of 67.7%, but the activity and stability of Lipase 2 were higher than those of Lipase 1. It is reported that amino acid composition plays a vital role in the determination of a protein’s structural conformation, which affects its flexibility and stability [58,59]. To investigate the cause of the high activity and stability of Lipase 2, the amino acid composition of Lipase 1 and Lipase 2 was analyzed. As shown in Appendix A, both Lipase 1 and Lipase 2 possess high glycine with 15.8% and 15%, and low cysteine with 1.4% and 1.3%, respectively, which is consistent with previous reports about cold-active lipases [53]. However, Lipase 2 possesses more Ser, Asn, Tyr, and Phe than does Lipase 1, which may explain why Lipase 2 is more stable than Lipase 1. Just as previously reported, polar residues like Ser and Asn can improve the stability of proteins by forming hydrogen bonds, and aromatic amino acids Tyr, Phe, and Trp also contribute to protein stability by aromatic interaction [60], but further experiments are still needed in the future.

## Figures and Tables

**Figure 1 microorganisms-09-00802-f001:**
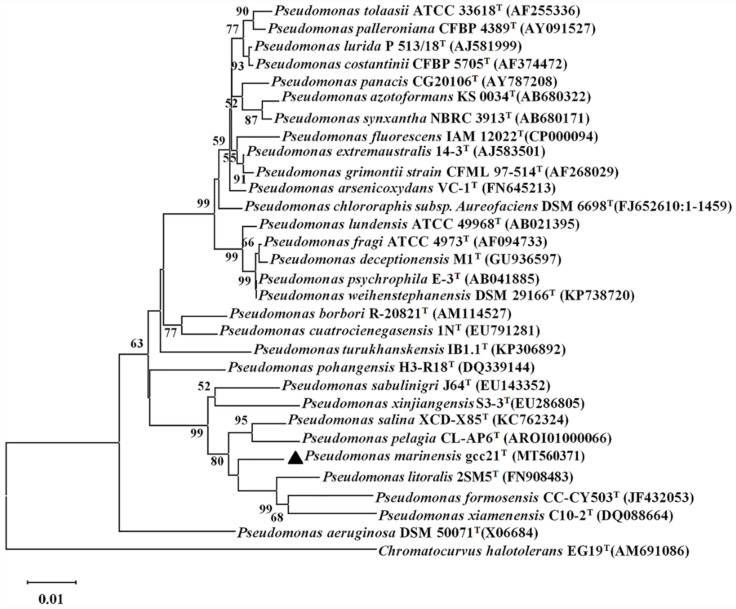
Neighbor-joining tree based on 16S rRNA gene sequences of *P. marinensis* gcc21^T^ and related taxa. The black triangle represents the evolutionary position of *P. marinensis* gcc21^T^. The 16S rRNA sequence of *Chromatocurvus halotolerans* EG19^T^ was used as an outgroup. Bootstrap values at nodes were derived from 1000 replicates. Only bootstrap values higher than 50% are shown. Bar, 0.01 substitutions per nucleotide position.

**Figure 2 microorganisms-09-00802-f002:**
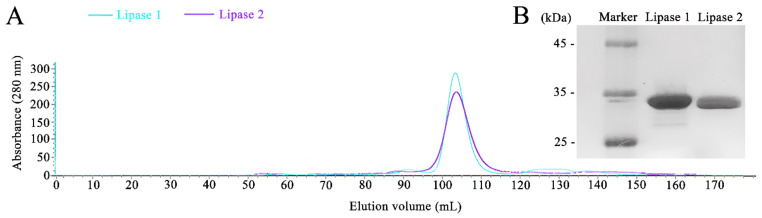
Purification of overexpressed Lipase 1 and Lipase 2 in *E. coli* cells. (**A**) Gel filtration chromatographies of Lipase 1 and Lipase 2 after purification by the Hiload^TM^ 16/600 Superdex^TM^ 200 column. (**B**) SDS-PAGE analysis of the purities of Lipase 1 and Lipase 2. Lane 1, standard molecular weight markers; lane 2, purified Lipase 1; lane 3, purified Lipase 2.

**Figure 3 microorganisms-09-00802-f003:**
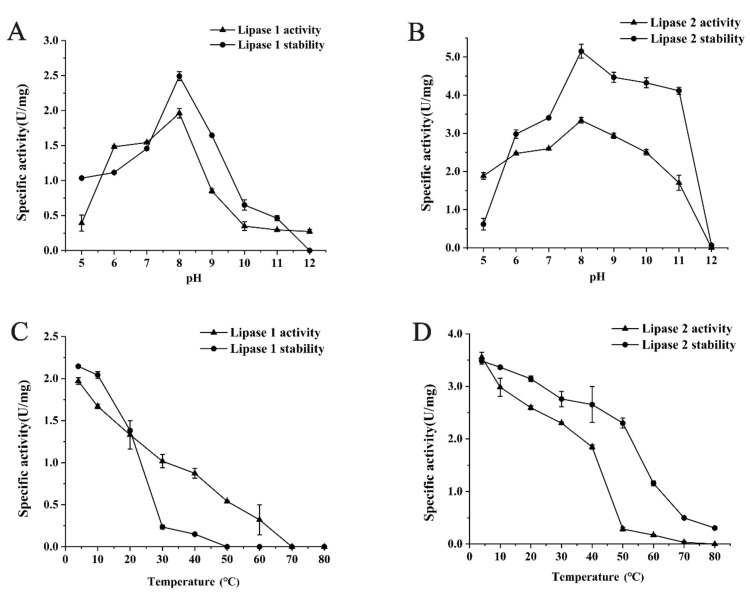
Effects of pH and temperature on the activity and stability of lipases. Effects of pH on the activity and stability of Lipase 1 (**A**) and Lipase 2 (**B**). Effects of temperature on the activity and stability of Lipase 1 (**C**) and Lipase 2 (**D**).

**Figure 4 microorganisms-09-00802-f004:**
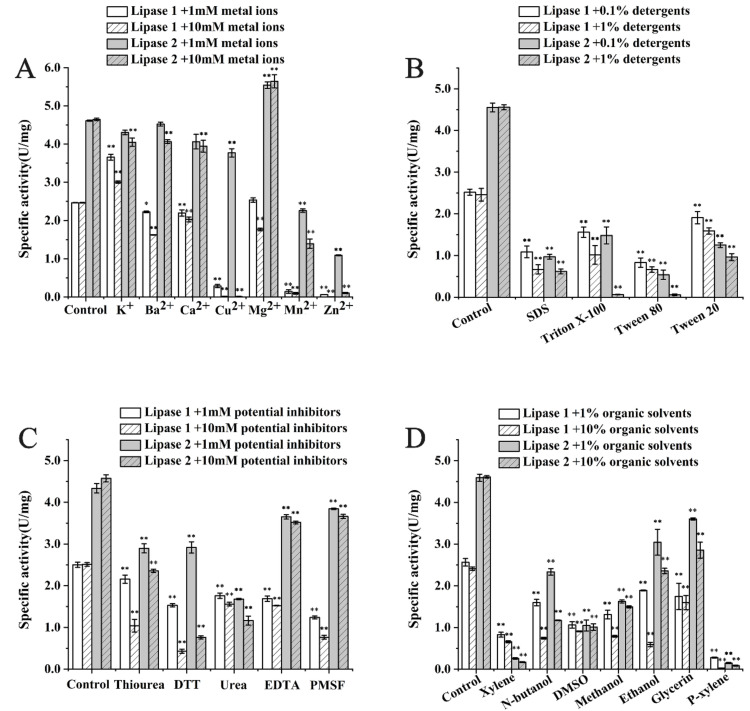
Stability of Lipase 1 and Lipase 2 against different metal ions (**A**), detergents (**B**), potential inhibitors (**C**), and organic solvents (**D**). All the data are represented by the mean ± standard deviation (SD). * *p* < 0.05, ** *p* < 0.01 vs. Control (The enzyme solution without any treatment).

**Figure 5 microorganisms-09-00802-f005:**
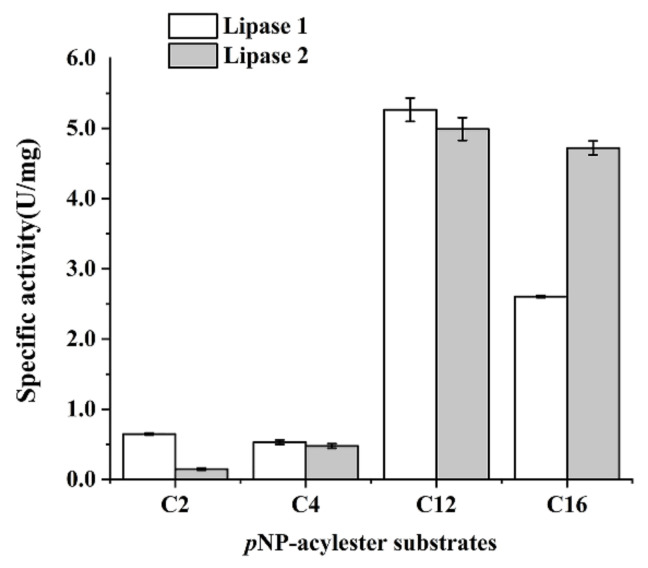
Effects of different substrates on the activities of Lipase 1 and Lipase 2. The final enzyme concentration in each reaction was 0.5 μg/mL. C2, *p*-nitrophenyl acetate; C4, *p*-nitrophenyl butyrate; C12, *p*-nitrophenyl laurate; C16, *p*-nitrophenyl palmitate.

**Figure 6 microorganisms-09-00802-f006:**
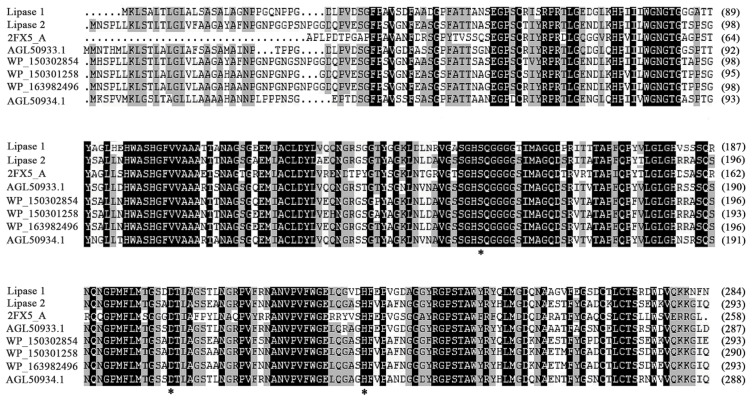
Alignment of Lipase 1, Lipase 2, and other reported homologs, in which 2FX5_A indicates a lipase from *Pseudomonas mendocina*, AGL50933.1 indicates a cold-active lipase precursor from an uncultured bacterium, WP_150302854 indicates an alpha/beta hydrolase from *Pseudomonas* sp. 16W4-4-3^T^, WP_150301258 indicates an alpha/beta hydrolase from *Pseudomonas profundi*, WP_163982496 indicates an alpha/beta hydrolase from *Pseudomonas* sp. OIL-1^T^, and AGL50934.1 indicates a cold-active lipase precursor from an uncultured bacterium. Asterisks indicate the mutation sites.

**Figure 7 microorganisms-09-00802-f007:**
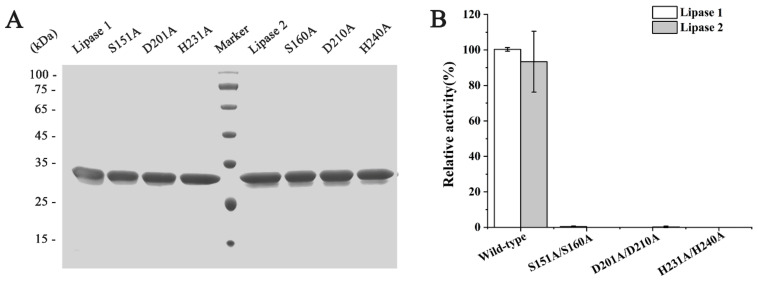
Activity assays of wild type and different mutants of Lipase 1 and Lipase 2. (**A**) Purification detection of wild type and corresponding mutants of Lipase 1 and Lipase 2 through SDS-PAGE. (**B**) Activity detection of wild type and corresponding mutants of Lipase 1 and Lipase 2, in which the activities of wild type Lipase 1 and Lipase 2 were defined as 100%.

**Figure 8 microorganisms-09-00802-f008:**
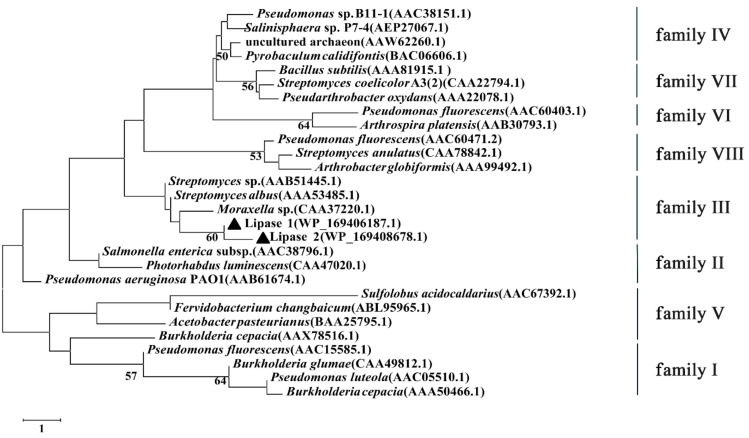
Phylogenetic tree of Lipase 1 and Lipase 2 based on conserved amino acid sequences. The black triangle represents the evolutionary position of Lipase 1 and Lipase 2. Bootstrap values at nodes were derived from 1000 replicates. Only bootstrap values higher than 50% were shown. Bar, 1 substitution per nucleotide position.

**Table 1 microorganisms-09-00802-t001:** Primers used in this study.

Name	Sequence (5′–3′)
27-F	AGAGTTTGATCCTGGCTCAG
1492-R	GGTTACCTTGTTACGACTT
Lipase 1-F	CGC**GAATTC**ATGAAGCTTTCCGCTCTT
Lipase 1-R	CCG**CTCGAG**GTTGAAATTCTTCTTCTG
Lipase 1-S151A-F	GCTCAAGGCGGTGGCGGCACCAT
Lipase 1-S151A-R	ATGGCCGGATGCGCCGACACGGT
Lipase 1-D201A-F	GCTACGCTGGCAGGCTCGACATT
Lipase 1-D201A-R	GTCGCTGCCGGTCATGAGGAAC
Lipase 1-H231A-F	GCTTTCGAACCCGTGGGCGA
Lipase 1-H231A-R	ATCGACTCCCTGCAGCTCACCC
Lipase 2-F	CGC**GGATCC**ATGAATTCACCCCTATTG
Lipase 2-R	CCC**AAGCTT**CTGGATGCCCTTTTTCTG
Lipase 2-S160A-F	GCTCAGGGCGGTGGCGGCTC
Lipase 2-S160A-R	GTGACCGGAGGAACCAACAGCGTCCAG
Lipase 2-D210A-F	GCCACCCTGGCCAGCTCCGA
Lipase 2-D210A-R	GGCGCTGCCAGTCATCAGGA
Lipase 2-H240A-F	GCCTTTGTTCCGGCCTTCAACGG
Lipase 2-H240A-R	GCTGGCCCCCTGCAACTCG

The nucleotides with bold and underlined characters represent the restriction sites.

**Table 2 microorganisms-09-00802-t002:** Characteristics comparison between strain gcc21^T^ and its related strains *^a^*.

Characteristics	1	2	3
**Cell morphology and size**	rod-shaped 0.8–1.0 µm × 1.1–1.8 µm	short-rod-shaped*^b^* 0.7–1.0 µm × 1.5–2.0 µm*^b^*	ND ND
**Growth condition**			
Temp (°C) for growth (optimal)	4–37 (28)	4–37 (28)	4–45 (28)
pH for growth (optimal)	5.0–8.5 (7.0)	5.5–9.0 (6.0)	4.5–9.0 (6.0)
NaCl concentration (%) for growth (optimal)	0.5–9.0 (1.5)	0–10.0 (7.0)	0–7.0 (2.0)
**Enzyme activity**			
Oxidase activity	+	-	+
Catalase activity	+	+	+
**Hydrolysis of**			
Starch	-	-	+
Tween 20	+	+	+
Tween 80	+	+	+
**Sole carbon source utilization**			
Ethanol	+	-	+
Sorbitol	+	+	-
D-mannose	+	+	-
L-arabitol	+	-	+
Xylitol	+	+	+
Acetate	+	+	+
Lactate	+	+	+
**Polar lipids**	DPG, PG, PE, PL1-4, APL	ND	ND
**Major fatty acids**	Branched-C_16:0_, Branched-C_17:0_ cyclo, Summed Feature 3, Summed Feature 8	Branched-C_12:0_, Branched-C_16:0_, Summed Feature 3, Summed Feature 8	Branched-C_19:0_ cyclo *ω*8 *c*, Branched-C_16:0_, Summed Feature 8
**DNA G+C content (%)**	58.27	58.1 *^b^*	66.6 *^c^*

Strains: 1, gcc21^T^ (all data from this study); 2, *Pseudomonas sabulinigri* J64^T^ (all data from this study except cell morphology and size, polar lipids, and DNA G+C content); 3, *Pseudomonas aeruginosa* DSM50071^T^ (all data from this study except cell morphology and size, polar lipids, and DNA G+C content). Summed features are groups of two or three fatty acids that could not be separated by GLC using the MIDI system. Summed feature 3 contains C18:1ω7c and/or C18:1ω6c, C16:1w7c and/or C16:1w6c, C16:1w6c and/or C16:1w7c. Summed feature 8 contains C18:1w7c, C18:1w6c. +, Positive result or growth; -, negative result or no growth. ND, not determined. *^a^* All data from this study except DNA G+C content and polar lipids. *^b^* Data from Kim K H et al. [39]. *^c^* Data from Mario von Neubeck et al. [40].

**Table 3 microorganisms-09-00802-t003:** Kinetics of *p*-NPP hydrolysis by Lipase 1 and Lipase 2.

Lipase	K_m_ (mM)	V_max_ (U/mg• S^−1^)	K_cat_ (S^−1^)	K_cat_/K_m_
Lipase 1	0.06	33.2	1.1	36.6
Lipase 2	0.02	154.8	2.7	140.1

For determination of kinetic parameters, substrate concentration ranges of 0.2–1.2 mM were used.

## Data Availability

The data presented in this study are openly available in NCBI GenBank, reference numbers (CP051625 and MT560371) for the genome and 16S rRNA gene sequence of *P. marinensis* gcc21, respectively, the reference numbers (WP_169406187.1 and WP_169408678.1) for the complete amino acid sequences of the Lipase 1 and Lipase 2, respectively, and the reference numbers (MW822015 and MW822016) for the corresponding DNA sequences of Lipase 1 and Lipase 2, respectively.

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
