# Peer review of "Characterization of Two Unique Cold-Active Lipases Derived from a Novel Deep-Sea Cold Seep Bacterium"

_microorganisms, 2021, doi:10.3390/microorganisms9040802_

Round 1

Reviewer 1 Report

The paper is a real interestig contribute in the bioprospecting field. English form requires extensive revisions. The paper is well structured but some details should be adjusted. A great effort is necessary in the valorization of the paper, which needs to be perfected. A deeper contextualization in the discussion, a correct report of methods and results should be addressed. Below my suggestions:

Lines 32-33. I think is better to express in plural form, as cold environments

Line 49. maybe widely?

Line 58. including Antarctica

Line 60. detergent production?

Line 75. Why did you choose a temperature incubation of 28°C if the strains were isolated from cold environment?

Lines 72-79. It is not clear if bacterial strains have been isolated in the framework of this paper or not. If yes, please provide details of isolation procedure and strains mainteinance.

Lines 111-116. Extraction and PCR reaction mixture description are absent.

Line 198. As I mentioned before, if the isolation was performed for this paper, here the authors should provide details on isolation results. And results from screening for liase-producing strains should be provided for all 250 isolated strains. 

Lines 262-264. These details should be moved in methods.

Line 276. Why obviously?

Figure 3. Please improve resolution

Line 290. I suggest to use anothe terms instead of could promote. It is a results not an hypothesis. Maybe could be better 'showed increase of lipase activity...' or something like that.

Lines 343-344- This is the results section, so no references should be added. Please, move in the discussion

In the Discussion, the contextualization of the work is completely absent. Moreover, you should discuss the isolation and the screening of a lot of strains, mentioned only at the beginning of the results. Even if you choose to go deeply inside with the most promising one, you should discuss  the efficiency rate of screening procedure on a high number of isolates. The results concerning the characterization of the strain object of the study, and that support it belongs to a new species should be underlined. Finally, results on the stability and specificity of lipases should be enphatized and better discussed. It is not enough to report the differences with previous work. It is important to understand how this manuscript contributes to the existing knowledge in the field.

Author Response

Thanks for the Reviewer's valuable advice, and we have modified our manuscript according to the Reviewer's  helpful suggestion. Please find our responses to Reviewer 1 in  attachment.

Reviewer 2 Report

The authors provided original research article entitled: "Characterization of two unique cold-active lipases derived from a novel deep-sea cold seep bacterium". The manuscript is very well written and below I present some minor suggestions:

Page 2, line 49 – „widely” instead of „wildly”

The aim of the work should be rephrased and clearly presented.

“Statistical analysis” subsection was omitted in the materials and methods part. Figure 4 may be statistically analyzed. ANOVA and post-hoc test may be performed.

Fig S1 – change the caption of the figure.

“ANIb, ANIm and isDDH” should be defined in the table footer in Table S2

Figures 3, 4, 5, and 7 are of low quality. Please improve it.

A more in-depth discussion should be provided.

Author Response

Thanks for the Reviewer's valuable advice, and we have modified our manuscript according to the Reviewer's  helpful suggestion. Please find our responses to Reviewer 2 in  attachment.

Reviewer 3 Report

The authors isolated a bacterial strain from deep-sea sediment. This strain produces two cold-active lipases. The authors identified and characterized the strain and the lipases.

This manuscript addresses an interesting subject. Moreover, the manuscript is well conducted and easy to understand, even if some sentences should be modified.

A major note: some parts of “Materials and methods” or “Results” have to be more detailed for the paper to be accepted. I wrote “very important note”.

Abstract:

Line 12: “producing” should be replaced by “produces”.

Line 13: “characterizations” should be replaced by “characteristics”.

Line 28: “producing” should be replaced by “produces”.

Introduction:

Line 32-33: I suggest “Cold environment such as the deep-sea [1], glaciers and mountain regions, is one of the most abundant environments in microorganisms on the earth’s surface“ instead of “Cold environment is one of the most abundant environment on the earth’s surface, such as the deep-sea [1], glaciers and mountain regions.”

Line 38-39: I suggest “Cold-active enzymes usually have high catalytic activities at temperatures below 25 °C, which gives them great advantages in detergent, textile and food industries because of energy savings they provide [4]”.

Line 49: “widely” instead of “wildly”.

Line 52: I suggest “due to the easy culture of microorganisms”.

Line 58: “including” instead of “include”.

Line 66 to 69: I suggest “named Pseudomonas marinensis gcc21 according to the results of genome sequencing and biological characteristics. Furthermore, the genes of strain P. marinensis gcc21 encoding two cold-active lipases were cloned and overexpressed in Escherichia coli cells, then purified and biochemically characterized”.

Materials and methods:

From the end of line 90 to line 101: the methods should be more detailed, or references of papers describing these methods should be added, in order to allow the reader to do these experiments for his/her own research.

Line 102: I suggest “The cultures were incubated at 28 °C in a shaker with a speed of 150 rpm for 5 days, and the absorbance was measured at 600 nm (A600)”.

Paragraph “2.3. Phylogenetic analysis”: The sequencing technique could be quoted.

Very important note: Line 126: The PCR conditions should be described. Moreover, “in silico”, I could not anneal the Lipase primers of Table 1, neither on the genome of the gcc21 strain nor on lipase genes, and theoretical annealing temperatures are very very high. Are the primer sequences correct?

Results

Very important note: Line 127, the protocol should be more precise. Which restriction enzyme(s) was(were) used? The primers and the pET-28a(+) vector do not seem to be compatible.

Line 128: “respectivity” should be removed.

Line 131: the sequencing technique could be indicated.

Lines 131-134: the term "recombinants" is not clear. Is it the plasmid ("transferred into E. coli") or the strain (“incubated in Luria-Bertani medium”)?

Line 138: “medium” should be removed.

Line 139: “diluted” should be replaced by “resuspended”.

Line 147: A dot (between “filtration” and “after”) could separate 2 sentences.

Line 163-164: A dot (between the last word of line 163, “min” and “absorbance”) could separate 2 sentences.

Line 182: A dot (between “above” and “each”) could separate 2 sentences.

Line 192-193: The Lipase sequences are amino-acid sequences. It could be interesting to deposit the DNA sequence of both genes in GenBank.

Lines 205-209: I suggest “Strain gcc21 was a Gram-negative, rod-shaped bacterium, 0.8 to 1.0 µm wide, 1.1 to 1.8 µm long, as shown by transmission electronic microscopy (TEM) (Figure S1). As indicated in Table 2, the strain was able to grow at a temperature of 4 to 37 °C (optimum, 28 °C), a pH ranging from 5.0 to 8.5 (optimum, pH 7.0), and a NaCl concentration of 0.5 to 9.0% (wt/vol) (optimum, 1.5%).”

Important note: Line 207: It is surprising that the optimum growth temperature is 28°C, as the strain was isolated from deep-sea cold seep. Do you know the temperature of this cold seep? It is surprising also that strain gcc21 could not use glucose (line 212). These results should be discussed.

Line 220: a “s” should be added to “amount”

Line 259 and 264: “respectively” could be removed

Lines 275 to 284: I would simplify. I propose to write: Lipase 1 showed an obvious activity over a wide pH range, from 5.0 to 11.0, with the highest activity at the pH 8.0. This Lipase also exhibited the highest stability at pH  8.0 (Figure 3A). Lipase 2 exhibited similar optimal pH activity and stability, but showed higher activity and stability (Figure 3B). Lipase 1 exhibited the highest activity and stability at the lowest temperature tested (4 °C), while the activity and stability decreased sharply with increasing temperature (Figure 3C). Lipase 2 showed similar activity and stability, however, gradually decreasing with increasing temperature. (Figure 3D).   

Important note: Figure 3: The activity and stability curves of lipases cannot be differentiated.

Line 289: “respectively” should be removed and “detected” could be replaced by “studied” or “assayed”.

Line 295: I would write “reduced” instead of “inhibited”

Line 296: I would end the sentence with a dot after (figure 4C).

Line 297: “detection” could be replaced by “assays”.

Important note: Are the differences observed in Figure 4 significant? It would be interesting to give statistical information (Student test or ANOVA).

Line 302: I would remove “detection”. It could be interesting to remind what is the control.

Line 305: Why did you test the lipase activity at room temperature (and not at 4°C for example)?

Line 309: A dot (after ”Lipase 2”) could separate 2 sentences.

Line 316: A “s” should be added to “Kinetic”.

Line 365: I am not sure that 4°C is the optimal temperature. It could be a lower temperature. “which is similar to” could be replaced by “as for”.

Discussion

Line 366: A dot after “[43]” could separate 2 sentences.

Line 376: You should write “high” instead of “highly”.

Line 385: I would simplify the sentence (I propose “the activity and stability of Lipase 2 are higher than those of Lipase 1”).

Line 390: “contents of”, “those of” could be removed. 

Author Response

Thanks for the Reviewer's valuable advice, and we have modified our manuscript according to the Reviewer's  helpful suggestion. Please find our responses to Reviewer 3 in  attachment.
